# Tracking Immature Testicular Tissue after Vitrification In Vitro and In Vivo for Pre-Pubertal Fertility Preservation: A Translational Transgenic Mouse Model

**DOI:** 10.3390/ijms23158425

**Published:** 2022-07-29

**Authors:** Buo-Jia Lu, Ya-Li Huang, Yung-Liang Liu, Brian Shiian Chen, Bou-Zenn Lin, Chi-Huang Chen

**Affiliations:** 1Division of Reproductive Medicine, Department of Obstetrics and Gynecology, Taipei Medical University Hospital, Taipei 11031, Taiwan; beckcha050121@gmail.com; 2Department of Public Health, School of Medicine, College of Medicine, Taipei Medical University, Taipei 11031, Taiwan; ylhuang@tmu.edu.tw; 3Division of Infertility, Department of Obstetrics and Gynecology, Chung Shan Medical University Hospital, Taichung 40201, Taiwan; h121976@yahoo.com.tw; 4School of Medicine, Chung Shan Medical University, Taichung 40201, Taiwan; 1638faang@gmail.com; 5Department of Gastroenterology, Ren-Ai Branch, Taipei City Hospital, Taipei 11217, Taiwan; td00268522@hotmail.com; 6Department of Obstetrics and Gynecology, School of Medicine, College of Medicine, Taipei Medical University, Taipei 11031, Taiwan

**Keywords:** pre-pubertal, fertility preservation, immature testicular tissue (ITT), solid surface vitrification (SSV), bioluminescence imaging (BLI), quantum yield (QY)

## Abstract

Pediatric cancer survivors experiencing gonadotoxic chemoradiation therapy may encounter subfertility or permanent infertility. However, previous studies of cryopreservation of immature testicular tissue (ITT) have mainly been limited to in vitro studies. In this study, we aim to evaluate in vitro and in vivo bioluminescence imaging (BLI) for solid surface-vitrified (SSV) ITT grafts until adulthood. The donors and recipients were transgenic and wild-type mice, respectively, with fresh ITT grafts used as the control group. In our study, the frozen ITT grafts remained intact as shown in the BLI, scanning electron microscopy (SEM) and immunohistochemistry (IHC) analyses. Graft survival was analyzed by BLI on days 1, 2, 5, 7, and 31 after transplantation. The signals decreased by quantum yield between days 2 and 5 in both groups, but gradually increased afterwards until day 31, which were significantly stronger than day 1 after transplantation (*p* = 0.008). The differences between the two groups were constantly insignificant, suggesting that both fresh and SSV ITT can survive, accompanied by spermatogenesis, until adulthood. The ITT in both groups presented similar BLI intensity and intact cells and ultrastructures for spermatogenesis. This translational model demonstrates the great potential of SSV for ITT in pre-pubertal male fertility preservation.

## 1. Introduction

According to cancer statistics in 2020, the incidence of childhood and adolescent cancer has increased slightly, while overall mortality has greatly reduced, declining by 68% in children and 63% in adolescents [1]. As many cancer therapies are considered gonadotoxic, fertility preservation has become an important issue for these surviving patients [2].

Guidelines published by the American Society of Clinical Oncology in 2018 indicate that, for pre-pubertal children, the only fertility preservation option is cryopreservation of ovarian or testicular tissue using methods that are currently under investigation [3]. The first and only live birth after an autograft of ovarian tissue cryopreserved during childhood was reported in 2015 and 2017 [4,5]. For males, previous studies on rhesus monkeys demonstrated that autologous grafting of cryopreserved pre-pubertal testis could produce sperm and healthy offspring [6]. However, to our knowledge, there has been no live birth following cryopreserved immature testicular tissue (ITT) transplantation in humans.

The optimal method for ITT cryopreservation remains unknown and there is no standardized protocol [2]. There are two methods for ITT cryopreservation, slow freezing and vitrification. Currently, controlled slow freezing (CSF) is the most commonly used freezing protocol globally [7]. Vitrification is a faster and cheaper alternative and may avoid ice crystal formation and the ensuing injury to the tissues [8]. Early vitrification methods that applied a variety of devices, including electron microscope grids [9], open pulled straws [10], nylon loops [11], solid surfaces [12], and dropping samples directly into liquid nitrogen (LN_2_) [13], have been used for oocytes, embryos, and ovarian tissues.

To date, direct vitrification in cryovials is widely used for oocyte and embryo cryopreservation, while solid surface vitrification (SSV) has previously been adopted in many studies for testicular tissue. Most of these studies evaluated the tissue by in vitro morphology and immunohistochemistry (IHC) assessment rather than by the function of the tissue transplanted in vivo [14,15,16,17,18]. Two studies comparing CSF and vitrification for human ITT cryopreservation have demonstrated that vitrification is able to maintain the proliferation capacity of germ cells and may be a feasible alternative [8,17]. Increasingly, the trend has been to use SSV for gonadal tissue as exemplified by my work on ovary preservation [19]. No previous studies have examined SSV with immature tissue grafts from pre-pubertal to adulthood both in vitro and in vivo as demonstrated herein.

The development of noninvasive bioluminescence imaging (BLI) makes it possible to follow tissue-specific luciferase expression in transgenic mice and monitor biological processes, such as signaling or protein–protein interactions of transplanted tissues in vivo [20]. This technology has been applied in our experiments to track germ cells, ovarian tissues, and ITTs for over a dozen years. As described in our previous serial studies [21,22], we used FVB/N-Tg (*PolII-luc*) Ltc transgenic mice and showed that BLI is a viable tool to track the survival of ovarian grafts. Later, BLI was introduced as a tool to evaluate germ cells and ITT in pre-pubertal male mice [23,24]. 

Testicular cryopreservation is still considered an experimental procedure and the protocols vary between studies. More studies are necessary not only to optimize the protocol, but also to make this technology more patient-friendly. The purpose of this study is to investigate the feasibility of SSV for ITT cryopreservation. We studied the ITT in vitro by BLI, scanning electron microscopy (SEM) [25] and ultrastructure of seminiferous tubules by IHC after SSV and rewarming, and the histology of ITT grafts 31 days after transplantation. More importantly, we used BLI to monitor the in vivo fate of ITT grafts longitudinally to determine their biological activity and efficacy in a real-time manner.

This transgenic translational study provides a surrogate model for a more comprehensive way to evaluate SSV/rewarmed ITT both in vitro and in vivo by BLI technology until adulthood.

## 2. Results

### 2.1. Tracking ITT Grafts after SSV and Rewarming In Vitro and In Vivo until Adulthood for Pre-Pubertal Fertility Preservation

The study design was based on our study published recently by *IJMS* [24]. The main focus was on ITT received from weaned 3-week-old FVB/N-Tg (*PolII-luc*) Ltc transgenic donor mice processed into tiny fragments, followed by tracking the frozen/thawed ITT grafts. For each age-matched donor, FVB/N-Tg mice underwent orchiectomy sparing the scrotum as the donor tissues. Briefly, the equilibrated ITT fragments were treated with cryoprotectant agent (CPA) and swiftly dropped as a microdrop (~30 μL) onto the top of a metal cube partially immersed in LN_2_. After rewarming, the SSV ITT fragments were analyzed for the ultrastructure by IHC and in vitro BLI in a 96-well plate.

The recipient underwent unilateral orchiectomy and 10 μL of ITT fragments were transplanted. BLI was applied as a tool to track the donor graft over time for 31 days until adulthood (Figure 1).

### 2.2. ITT Fragments of 3-Week-Old FVB/N-Tg (PolII-luc) Ltc Mice Used as Donor Grafts

Donor mice were anesthetized before an incision exposed the testis from the scrotum (Figure 2A). Each testis measured about 4 × 3 × 3 mm^3^ (Figure 2B). We minced the ITT with a blade to tiny fragments that were washed in Dulbecco’s phosphate-buffered saline (DPBS; Thermo Fisher Scientific; Waltham, MA, USA). Figure 2C shows fragmented ITT tissues. The magnified figure indicates each fragment size was approximately 0.29 mm × 0.12 mm on average (Figure 2D). The figure shows multiple CPA droplets on a metal cube with partial LN_2_ immersion during the SSV procedure (Figure 2E).

### 2.3. In Vitro Quantity-Based BLI of ITT Fragments by Quantum Yields (QYs) in the Culture Plate

Noninvasive quantitative molecular imaging provided more imaging modalities, greater sensitivity by promoter *polII*, and modified firefly luciferase cDNA (pGL-2, Promega, Madison, WI, USA). We categorized the samples into five groups. The first group is DPBS buffer only without ITT tissues, the second group is 1 µL of fresh ITT, the third group is negative control, the fourth group is 5 µL vitrified ITT without CPA, and the fifth group is 5 µL vitrified ITT with CPA (Figure 3A). We only added 1 µL, instead of 5 µL, of fresh ITT because of the limitation of the saturated BLI image. The QYs for 5 µL of fresh ITT is too strong to be analyzed by this system. As a result, the QYs in the first, the third and the fourth group are very low. There is statistically significant higher QYs in the second group than that in the fifth group (*p* = 0.04) (Figure 3B).

### 2.4. The Ultrastructure of Fresh ITT, Vitrfied ITT without and with CPA by SEM

Images of the fragmented testes were taken with a S3500 Tabletop SEM (Hitachi, Tokyo, Japan) at 400× magnification at an accelerating voltage of 15 kV. Figure 4 demonstrates the ultrastructure for fresh ITT, vitrified ITT without CPA, and vitrified ITT with CPA. We marked the structures, including Leydig cell, Sertoli cell, spermatogonia and spermatocyte. The mild crease shrinkage was noted on the outer surface of seminiferous tubules in vitrified ITT with CPA (Figure 4C), which was about two-thirds the size of the one in the fresh ITT (Figure 4A). For vitrified ITT group without CPA, the outer surface showed significant crease-shrinkage compared with the other two groups, and the diameter of the seminiferous tubule was the smallest (Figure 4B).

### 2.5. The Ultrastructure of Seminiferous Tubules of Fresh and SSV Groups by IHC

The expression of cell markers, including Oct4 [26], Sox-9 [27], 3β-HSD [28], and DDX4 [29], showed an intact ultrastructure containing undifferentiated spermatogonia, Sertoli cells, Leydig cells, and germ cells (arrow), respectively, in both the fresh control and SSV group. Intact Leydig cells, Sertoli cells, and undifferentiated spermatogonia represent early spermatogenesis without mature sperm in both groups before transplantation (Figure 5).

### 2.6. Tracking the In Vivo BLI of Transgenic Mouse ITT Grafts in Fresh and SSV Group over Time

After we removed one of the recipients’ testes, we replaced it with fresh or SSV ITT from the FVB/N-Tg (*PolII-luc*) Ltc transgenic mice. We traced the survival of the graft by in vivo BLI on days 1, 2, 5, 7, and 31 after transplantation (Figure 6A). For both the vitrification and the control groups, the BLI QYs decreased between days 2 and 5. After day 5, the QYs showed an upward trend until day 31. Compared with day 1, the QYs was significantly stronger at day 31 after transplantation (9.3 × 10^5^ photons/s versus 15.4 × 10^5^ photons/s, *p* = 0.008). The signals were lower than the control group after day 7 (Figure 6B). However, the differences between the two groups failed to reach significance from the 1st day after transplantation until the 31st day. (*p* > 0.05) (Figure 6C). These results indicate that both fresh and SSV ITT can survive for at least 31 days until adulthood after transplantation. Moreover, the vitrification group revealed BLI signal intensity that was comparable with that of the fresh tissues.

### 2.7. Testicular Grafts of Hematoxylin and Eosin Staining

Slide images were taken with MoticEasyScan Pro 6, and morphological changes in the testicular structures were examined with Motic DSAssistant software. The gradual magnification of testicular tissue containing intact connection seminiferous tubules with epididymis showed progressive spermatogenesis with germ cells, especially mature sperm marked in the red circle in fresh grafts (Figure 7A) and SSV grafts (Figure 7B).

## 3. Discussion

Many studies have investigated the feasibility of vitrification for ITT cryopreservation. To our knowledge, this is the first study to report the survival of ITT in vivo by adopting the BLI imaging system. After SSV and rewarming, we found that the BLI intensity was significantly higher in the fresh group than in the vitrified ITT with CPA. The IHC stain showed that the vitrified tissues preserved the structures and cells for spermatogenesis. During longitudinal observation of the bioluminescence in vivo for 31 days, we confirmed that the tissues were viable after transplantation in both the SSV and the control groups with comparable BLI intensity. The intensity was significantly stronger on day 31 compared with the first day, indicating the possibility of tissue self-repair after transplantation to achieve progressive spermatogenesis. On day 31, intact seminiferous tubules and mature sperms were noted in both SSV and fresh control group.

The aim of our study was to adopt the translational model of BLI signals in transgenic mice to track the survival of the grafts in vivo. In our previous study, this animal model was shown to be useful for quantifying germ cells in vitro and assessing the efficacy of germ cell transplantation in vivo [23]. In the present study, BLI signals were initially stronger in the fresh group than in the SSV group in vitro. At follow-up on days 1, 2, 5, 7, and 31 after transplantation, BLI intensities were similar between the SSV and the control group. This finding suggests that ITT can recover from the vitrification–rewarming process and establish revascularization in a manner similar to fresh tissues from the beginning of transplantation. This observation is comparable with earlier in vitro studies that reported similar outcomes with vitrification or the fresh controls [17,30,31]. Unlike in ITT, ovarian cryopreservation by slow freezing may compromise ovarian reserve through cryoinjury and ischemia. The process of revascularization takes 2–7 days to complete, depending on the size of the implant [32]. The bioluminescence intensity decreased between days 2 and 5, and then increased linearly. The BLI signals were significantly higher on day 31 than on day 1, indicating the presence of revascularization after day 5, and the subsequent restoration of function of the ITT grafts.

The *PolII-luc* transgenic animal model may be used as donor animals for studying transplantation, lineage tracing, and other biomedical processes. Luciferase expression is detectable in all organs, with the highest expression in skin, pancreas, and testis according to the manufacturer’s instruction. BLI both in vitro and in vivo quantifies and correlates experimental models in all imaging modes. Working with bioluminescent and fluorescent reporters in the same subject facilitates changing imaging modes. The IVIS Spectrum handles it all, from microplates to high-throughput whole animal studies. With this system, we have published a novel study investigating the application of poly-L-lactic acid (PLLA) scaffold in age-matched donors and recipients by adopting this *PolII-luc* transgenic animal model [24]. The result showed that the grafted fresh testicular tissues with scaffold could significantly increase the degree of sperm proliferation compared with those without the scaffold, in particular, on the 5th, 7th, and 42nd days after transplantation. Previous studies have reported that PLLA increased the in vitro cluster formation of neonate fresh and frozen–thawed spermatogonial cells and caused them to differentiate during cultivation [33]. The advantage of this study is that an animal model was established to evaluate the survival of ITT both in vivo and in vitro. In a future study, we will explore the application of PLLA in ITT after SSV by adopting this experimental design.

According to previous reviews, eight studies have compared vitrification with slow freezing for ITT cryo-storage [8,14,15,16,17,18,34,35]. In 2010, Abrishami et al. reported that ITT vitrification could maintain cell viability and restore spermatogenesis after a xenograft [14]. They found that exposure to dimethyl sulfoxide (DMSO) for 5 min yielded numerically higher cell numbers than in grafts exposed for 15 or 30 min. Accordingly, we limited DMSO exposure of our grafts to less than 10 min. Curaba et al. compared vitrification and slow freezing in terms of IHC and tissue integrity. They confirmed that vitrification was a promising approach, but additional studies should be conducted in vivo to assess the completion of spermatogenesis [15]. The same authors also published a case report assessing the vitrification of human ITT and revealed that the histology characteristics of spermatogonia and Sertoli cells were preserved [8]. Gouk et al. focused on harvesting spermatogonial stem cells (SSCs) for fertility preservation. They found that vitrification maintained post-warming cell viability and function significantly better than conventional slow and rapid freezing protocols [35]. Baert et al. and Poels et al. both agreed that vitrification is an effective strategy to maintain the proliferative capacity of SSCs and integrity of ITT [17,34]. Poels et al. were the first to adopt a human ITT xenotransplantation model. They observed spermatogonia proliferation 6 months after transplantation, but then there was a blockage at the pachytene stage. Time-consuming CSF protocols are commonly used in human testicular tissue banking. Baert et al. have investigated the alternatives to conventional CSF using testicular tissues from 14 adult patients. In the vitrification group, they found increased numbers of seminiferous tubules displaying a ruptured epithelium and considered this method to have a negative impact on spermatogonial number [18]. By contrast, Dumont et al. reported the superiority of the vitrification protocol in terms of testicular structure maintenance, tubular morphology, and tissue function [16]. Recent review concluded that the results of comparison between vitrification and slow freezing for ITT were still conflicting, and most of them were limited to in vitro animal studies [2]. We summarized the experimental method and conclusion of each article in the following Table 1. At the same time, we compared our study with previous studies and highlighted the novelty of our work, especially in vivo BLI evaluation.

There are scanty studies reporting live birth after ITT cryopreservation. In 2002, using sperm developed in the frozen–thawed transplants, the first live mouse offspring was born after in vitro micro-insemination [36]. Another more recent study retrieved post-meiotic spermatids 8–12 months after transplantation and successfully produced offspring from autologous grafting of cryopreserved pre-pubertal rhesus monkeys [6]. Both studies adopted slow freezing for ITT cryopreservation. Kaneko et al. successfully generated porcine offspring utilizing sperm from immature testicular tissues after vitrification and transplantation into nude mice [37]. Testicular xenografting animal was adopted in this study. There are some ethical and methodological obstacles that need to be overcome before this technology could be applied to human beings. In addition, in vitro culture is an interesting method to avoid the re-introduction of malignant cells to cancer patients, especially with hematological malignancies [38]. Sato et al. were the first group to show complete spermatogenesis from cryopreserved ITT after they were thawed and culture in vitro. Offspring was generated from spermatids and sperm produced in vitro by micro-insemination [39]. Since then, the cryopreservation of a mixed population of testicular cell suspensions, enriched in SSCs, has been envisaged as a simple and cost-effective alternative to preserve male fertility. There are some recent studies investigated the optimal method for SSCs preservation, including SSV [40,41]. To the best of our knowledge, there is currently no live birth of human after either frozen-thawed ITT transplantation or in vitro culture.

There are several limitations to this study. First, the BLI signals could represent tissue viability and its ability for spermatogenesis, but not the stages of cell differentiation. Second, potential criticisms may come from germ cell proliferation at a slow rate as early as 1.5 postnatal days. However, in order to accommodate the in vivo BLI tracking system, we could only adopt weaning donor mice at 3-week-old, which is equal to the pre-pubertal age with germ cell proliferation at a slow rate, functional maturation of Sertoli cells, and the production of functional Leydig cells [42]. Third, we only evaluated in vivo BLI on days 1, 2, 5, 7, and 31, and a longer follow up duration and more frequent data collections may provide us more information about the trend of BLI in vivo. Fourth, we did not report a live birth from the transplanted ITT. More studies are required to prove that, after SSV and rewarming, the transplanted ITTs are able to produce viable offspring after the adoption of this in vivo BLI system. Finally, we only evaluated the ITT grossly or by SEM, IHC, and BLI expression. Many recent studies investigated the genetic expression of mammalian tissue to better understand the mechanisms of their functions [43,44,45]. Further studies are needed to prove that the ITT tissues could perform normally after SSV/rewarming at a genetic level.

## 4. Materials and Methods

### 4.1. Bioethics and Animals

All procedures were reviewed and approved by the Animal Experimental Committee at the Taipei Medical University, in accordance with the Guiding Principles for the Care and Use of Laboratory Animals (LAC-101-0090).

The donors were 3-week-old immature male FVB/N-Tg (*PolII-Luc*) Ltc transgenic mice with an H-2 haplotype (H2q). These mice were created by the transgenic service of Level Biotechnology (New Taipei City, Taiwan), and the generation process was described in our previous studies [22,23,24]. In brief, after a pronuclear microinjection of the *PolII-Luc* transgene into the FVB/N embryos, they could encode a 712-bp mouse RNA polymerase II promoter (*PolII*) and a modified firefly luciferase complementary cDNA (pGL-2, Promega, WI, USA). The animals were hemizygotes and could express the transgene for luciferase (*Luc*) and transmit this gene to their offspring. The recipients were 3-week-old immature FVB/NJNarl wild-type male mice with an H2q. We obtained these mice from the National Laboratory Animal Center (Taipei City, Taiwan).

A total of 12 pairs of donor and recipient mice were included in our experiments. They were all bred in the animal house of Taipei Medical University under the temperature of 22–24 °C and 12/12 h light/dark regimen.

### 4.2. Study Design and ITT Collection

For the present study, 24 immature FVB/N-Tg (*PolII-luc*) Ltc transgenic donor male mice, 3-week-old postweaning, were divided into two groups comprising the ITT after SSV and fresh control in vitro, and in vivo BLI tracking the developmental potential on a quality and quantity basis until adulthood. Euthanasia was performed under Isoflurane (Abbott; Chicago, IL, USA) anesthesia by dislocation of the cervical spine. The hair around the testicles was shaved before the operation. Pairs of testicles were removed from the donor scrotum and placed in an iced Petri dish. The seminiferous tubules were separated from surrounding tissues and fragmented into pieces less than 1 mm and washed three times with DPBS. As fragments can be easily lost in each step, we mixed fragments derived from each individual together. Subsequent experiments were carried out with this fragmented testicular mixture.

### 4.3. ITT Transplantation after Solid Surface Vitrification and Rewarming of ITT

For the SSV procedures, we adopted a methodology previously described for mice ITT [14,16,18,34] with some modifications. Briefly, to achieve slow equilibrium, we used a plunger but removed the syringe and a 21G needle (TERUMO, Tokyo, Japan) to simply limit the flow rate to an average of 33.4 μL per s. Using the abovementioned device and a shaker, we gently introduced 15 mL of vitrification solution (VS) within 10 min. Finally, we made the VS consisting of Leibovitz’s L-15 Medium (Thermo Fisher Scientific; Waltham, MA, USA) supplemented with 10.6% DMSO, 0.1 mol/L sucrose (Sigma; St. Louis, MI, USA), 10% FBS (Biological Industries; Beit HaEmek, Israel), and 1% penicillin-streptomycin (Thermo Fisher Scientific; Waltham, MA, USA). We placed the ITT mixture solution into a centrifuge tube and then waited a few minutes until ITT precipitation.

The supernatant was removed, and the precipitate was divided into 30 μL and dropped on a metal cube (CoolRack^®^, Corning Inc.; Somerville, NY, USA) surface partially immersed in LN_2_ to form a microdroplet. We then transferred the vitrified droplet into pre-cooled cryovials (Biomate, Taipei, Taiwan) followed by storage in LN_2_.

A week later, the cryovials were exposed to room temperature (25 °C) for 30 s, and we poured out the ITT beads and immersed them into 38 °C DPBS containing 1 M sucrose. For the removal of cryoprotectants and to slowly achieve equilibration, we gently introduced 15 mL cold DPBS with slow shaking over ten minutes. We poured the ITT mixture into 15 mL centrifuge tubes and waited a few minutes until the ITT precipitated; then, we removed the supernatant and washed the precipitate three times with DPBS.

In the experimental group, the ITT from the donor mice was processed for SSV and rewarming before transplantation. For each age-matched donor, FVB/N wild-type mice underwent unilateral orchiectomy sparing the scrotum as the recipients. Each transplanted testicular graft was completed within 2 h after warming. For the other 12 mice in the fresh control group, the ITT was transplanted to the recipients immediately. The loading amount measured volume was 10 μL.

We monitored all the transplanted grafts in vivo using BLI technology for 31 days. According to previous study [42], spermatogenesis begins since 30th postnatal day (PND). The mice should reach its puberty during 35th to 55th PND depending on different strain. Our donor mice were 21st PND and we could expect that the spermatogenesis should turn faster after 31 days of transplantation (52nd PND). On day 31, we removed the grafts and evaluated them by hematoxylin and eosin (H&E) staining.

### 4.4. Assessment of Thawed ITT Recovery In Vitro

BLI was performed using the IVIS Lumina XRMS In Vivo Imaging System (PerkinElmer Corp.; Waltham, MA, USA). The samples were mixed with D-luciferin (L_8220, Biosynth Carbosynth; Rietlistrasse, Staad, Switzerland) 10 min before imaging and placed into a light-tight camera box on the stage of the imaging chamber. An overlay image (black and white picture) was taken with the aid of light inside the imaging chamber. Luminescence was quantified using Living Image software (version 3.0; Caliper Life Sciences; Waltham, MA, USA).

We observe the activity of fresh and SSV/warming fragmented ITT in vitro in a 96-well plate (Corning Inc.; Somerville, NY, USA). We divided the ITT into five groups; the first group is DPBS buffer only without ITT tissue, USAs, the second group is 1 µL of fresh ITT, the third group is negative control, the fourth group is 5 µL vitrified ITT without CPA, and the fifth group is 5 µL vitrified ITT with CPA. Then, we calculated the average BLI QY and presented as histogram.

### 4.5. Scanning Electron Microscopy of the Seminiferous Tubule Ultrastructure of Fresh ITT, ITT without CPA, and ITT with CPA

The images of the ITT were taken with a S3500 Tabletop scanning electron microscope (Hitachi, Tokyo, Japan) at 400× magnification.

### 4.6. Immunohistochemistry Staining

ITT fragments were collected, and the tissue sections were incubated in 0.01 M citrate buffer (pH 6.0) at 95 °C for 30 min. Then, the sections were washed three times in PBS (phosphate-buffered saline; Thermo Fisher Scientific; Waltham, MA, USA). Endogenous peroxidase was blocked with H_2_O_2_ for 10 min. Next, unspecific staining was blocked with 0.5% nonfat milk in PBS for 1 h. We used the following antibody probes: octamer-binding transcription factor 4 (OCT 4 A7920; ABclonal; New Taipei City, Taiwan), sex-determining region Y-box 9 (Sox 9 A19710; ABclonal; New Taipei City, Taiwan), 3-hydroxysteroid dehydrogenase (3ß-HSD A1823; ABclonal; New Taipei City, Taiwan), and DEAD (Asp–Glu–Ala–Asp)-box polypeptide-4 (DDX4 51042-1-AP; Proteintech; Rosemont IL, USA). The sections were placed in a humidified chamber overnight at 4 °C and then incubated for 15 min with rabbit/mouse horseradish peroxidase-labeled secondary antibody. After washing with PBS three times, the sections were mounted on glass slides and images were captured with MoticEasyScan Pro 6 and examined with Motic DSAssistant software (Motic China; Xiamen, China).

### 4.7. In Vivo Tracking the Development of Vitrified ITT Grafts until 31 Days after Transplantation

BLI was performed using the IVIS 200 imaging system (Xenogen Corp.; Alameda, CA, USA). The recipients were injected with D-luciferin (150 mg/kg, L_8220, Biosynth Carbosynth; Rietlistrasse, Staad, Switzerland) intra-peritoneally 10 min before imaging, anesthetized, and placed into a light-tight camera box on the stage of the imaging chamber. An overlay image (black and white picture) was taken with the aid of light inside the imaging chamber. Luminescence was quantified using Living Image software by summing pixel intensities within the region of interest, as described in previous studies [20].

### 4.8. Testicular Grafts of Hematoxylin and Eosin Staining

For H&E staining, the mouse testis was fixed in Bouin’s solution (Sigma; St. Louis, MI, USA). Then, the tissues were embedded in paraffin, and 5 μm thick sections were cut [46]. Slide images were taken with MoticEasyScan Pro 6, and morphological changes of the testicular structures were examined with Motic DSAssistant software.

### 4.9. Statistical Analysis

The SAS package (version 9.4; SAS Institute, Cary, NC, USA) was used for all data analyses. The Mann–Whitney U test was used to compare the photon yields between the fresh and SSV group. *p* < 0.05 (two-tailed) indicated statistical significance.

## 5. Conclusions

Based on our surrogate mouse model, the in vivo BLI intensities of SSV and fresh ITT grafts were similar, indicating that SSV ITT grafts can survive, accompanied by progressively faster rate of spermatogenesis, until adulthood. This model may serve as a platform for future versatile study designs and applications from bench to bedside for pre-pubertal male fertility preservation.

## Figures and Tables

**Figure 1 ijms-23-08425-f001:**
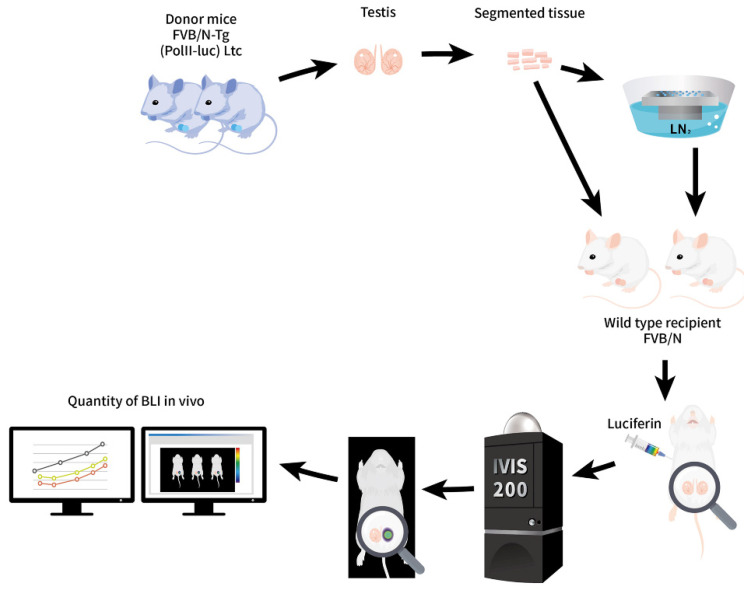
Schematic diagram of the study design. Isolated ITT from the FVB/N-Tg (*PolII-luc*) Ltc transgenic donor mice. Fragmented ITT were processed by SSV over LN_2_, followed by rewarming and iso-graft onto the scrotum of wild-type recipient mice that had undergone orchiectomy. For BLI, luciferin was injected before observing with an IVIS system. The amount of photon yield was proportional to the number of active live cells and tissues in vivo by digital data on a qualitative and quantitative basis. The BLI was tracked from 3-week-old until adulthood.

**Figure 2 ijms-23-08425-f002:**
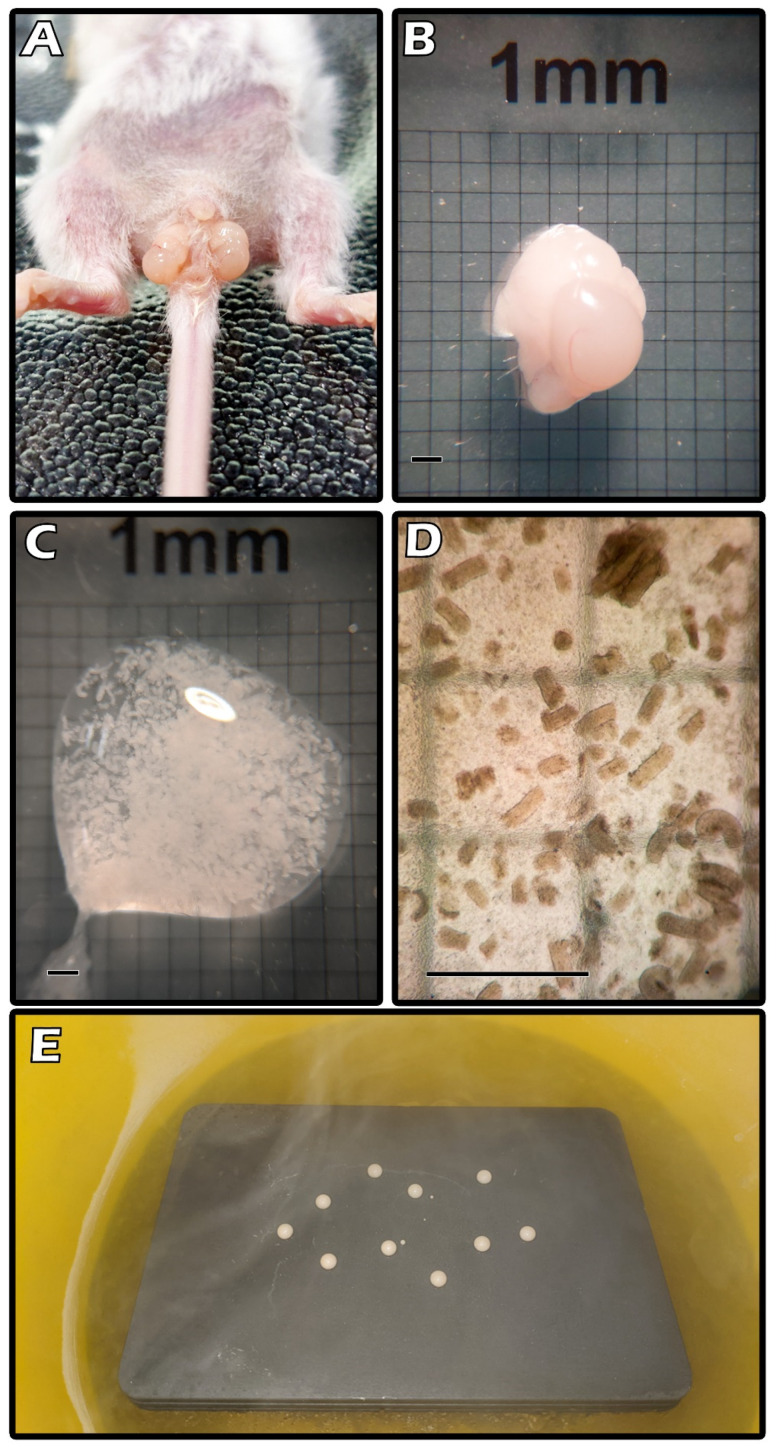
Exposed the testis from the scrotum (**A**). Each testis measured about 4 × 3 × 3 mm^3^ (**B**). Fragmented ITT tissues (**C**). Each fragment size was approximately 0.29 mm × 0.12 mm on average (**D**). Multiple CPA droplets on a metal cube with partial LN_2_ immersion during the SSV procedure (**E**).

**Figure 3 ijms-23-08425-f003:**
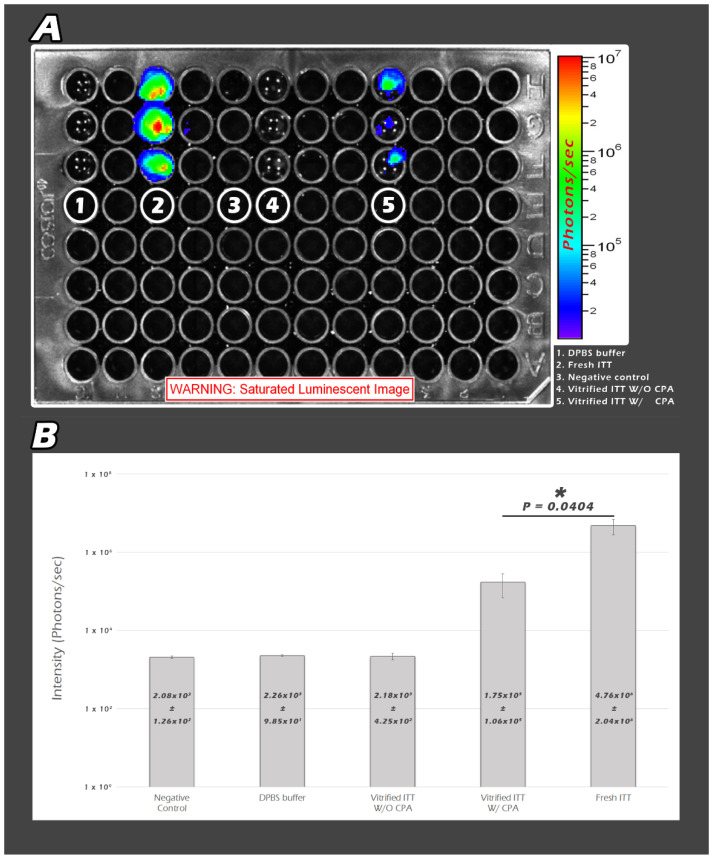
The second group (fresh ITT) demonstrated the highest signal. The fifth group (vitrified ITT W/CPA) is the second brightest. There were no visible BLI signals in the other three groups (**A**). Statistically significant higher QYs in the fresh ITT group than that in the vitrified ITT W/CPA group was noted (* *p* < 0.05) (**B**).

**Figure 4 ijms-23-08425-f004:**
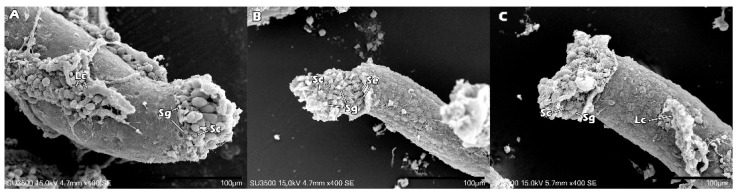
The mild crease shrinkage was noted on the outer surface of seminiferous tubules in the vitrified ITT with CPA (**C**), which was about two-thirds the size of seminiferous tubules of the fresh ITT (**A**). For vitrified ITT without CPA, the lumen of the seminiferous tubule was the smallest (**B**). Abbreviations: Lc, Leydig cell; Se, Sertoli cell; Sg, spermatogonia; and, Sc, spermatocyte.

**Figure 5 ijms-23-08425-f005:**
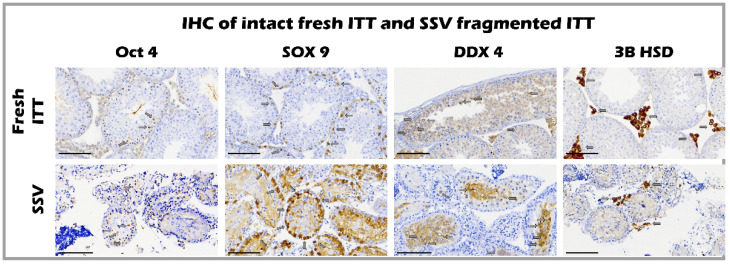
Expression of cell markers including Oct4, Sox-9, 3β-HSD, and DDX4 showed an intact ultrastructure containing undifferentiated spermatogonia, Sertoli cells, Leydig cells, and germ cells (arrow), respectively, in both fresh control and SSV group, bar = 100 μm.

**Figure 6 ijms-23-08425-f006:**
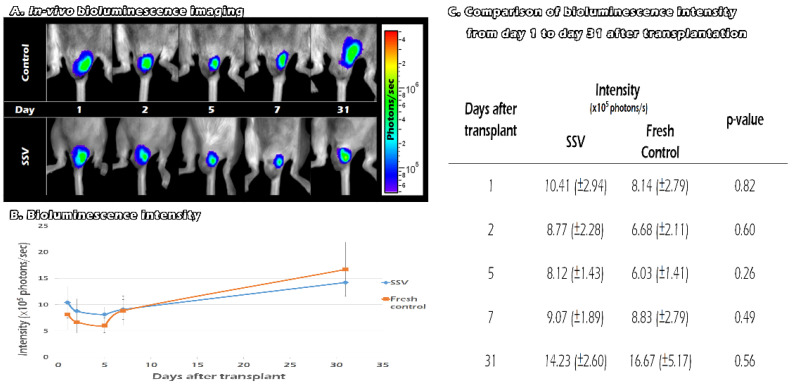
The survival of the graft by in vivo BLI on days 1, 2, 5, 7, and 31 after transplantation (**A**). Compared with day 1, the QYs was significantly stronger at day 31 after transplantation (9.3 × 10^5^ photons/s versus 15.4 × 10^5^ photons/s). The signals were lower than the control group after day 7 (**B**). There were no significant differences in the QYs between the two groups (**C**).

**Figure 7 ijms-23-08425-f007:**
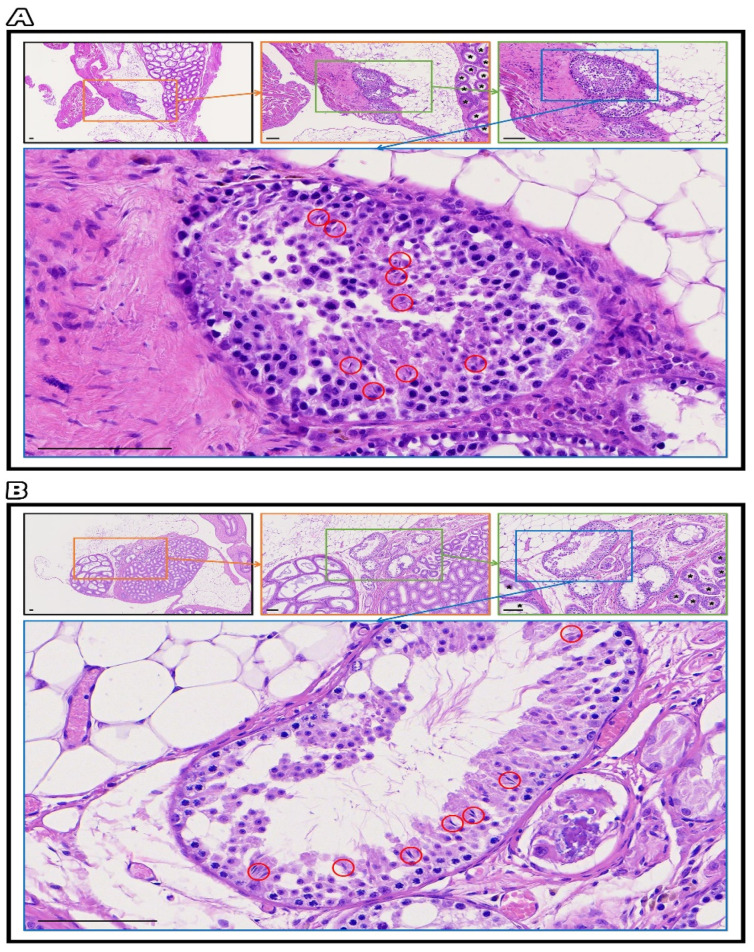
The gradual magnification of testicular tissue containing intact connection seminiferous tubules with epididymis showed progressive spermatogenesis with germ cells, especially mature sperm marked in the red circle in fresh grafts (**A**) and SSV grafts (**B**), bar = 100 μm.

**Table 1 ijms-23-08425-t001:** Summary of eight studies compared vitrification (V) with slow freezing (SF) for ITT cryo-storage.

Year	Author	Donor	Recipient/Culture	V Protocol	Evaluation	Conclusion
2010	Abrishami et al. [14]	Piglet	Mice	SSV	In vitro histology	SSV and SF presented normal spermatogenesis
2011	Curaba et al. [15]	Mice	In vitro short-term oganotypic culture	Cryo-straws	In vitro apoptotic marker and histology	V and SF preserve survival, development, and integrity of ITT
2011	Curaba et al. [8]	Human	10-day of organoty-pic culture	Cryo-straws	In vitro histology and IHC stain	V shows promise as an alternative strategy to SF
2011	Gouk et al. [35]	Mice	SSCs suspen-sions	Straw-in-straw method	In vitro evaluation and flow cytometry	V of testicular tissue is a time- and cost-efficient strategy to preserve SCC
2012	Baert et al. [34]	Mice	Mice	SSV	In vitro histology and IHC TEM	SSV resulted in success rates similar to SF
2013	Poels et al. [17]	Human	Mice	SSV	In vitro histology and IHC	SSV is able to maintain proliferation capacity in SSCs after 6 months of xenografting
2013	Baert et al. [18]	Huamn	No transplan-tation	SSV; Direct cover V	In vitro histology TEM	SSV may have a negative impact on spermatogonial number
2015	Dumont et al. [16]	Mice	In vitro culture	SSV; Open pulled straws; Single drop	In vitro histology and IHC Testosterone level Cell death assessment	SSV resulted in success rates better than SF for immediate frozen/thawed tissues but also after a long-term in vitro culture.
Our study 2022	Lu et al.	Trans-genic mice	Mice	SSV	In vitro histology; IHC; SEM In vivo and in vitro BLI imaging	SSV presented similar BLI intensity, intact cells and ultrastructures for spermatogenesis after transplantation with the fresh control

## Data Availability

All data generated in the study are presented in the manuscript.

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
