# Peer review of "Tracking Immature Testicular Tissue after Vitrification In Vitro and In Vivo for Pre-Pubertal Fertility Preservation: A Translational Transgenic Mouse Model"

_ijms, 2022, doi:10.3390/ijms23158425_

Round 1

Reviewer 1 Report

The manuscript by Buo-Jia Lu et al aimed at investigating the effect of Immature Testicular Tissue After Vitrification in Vitro and in Vivo for Prepubertal Fertility Preservation. The experimental part is complete, However, the article still needs major revisions to be accepted.

General Comment:

The objective of the manuscript is interesting, authors have performed a great work and the article will be interesting for readers. This is the first study to report the survival of ITT in vivo by adopting the BLI imaging system. This is a novel technique, which may provide a new idea for future research on testicular tissue cryopreservation.

General improvement:

1 The introductory part is too long, especially on Lines 42 and 79, the two sentences are probably the same meaning. It is recommended that the author shorten the whole introductory part. And in line 67-78, the author has quoted his 5 articles in this small paragraph, so it is recommended to reduce some of them.

2 The grouping of this article is still a bit confusing. Why are SSV group in Figure 3/5 parts divided into two groups (with CPA and without CPA), but not in Figure 4/6/7 parts?

3 About Figure 6 part, From the 7th to the 31st day, although bioluminescence intensity will increase, it is recommended to measure once or twice during this period.

4  It is recommended to revise the methods and materials part as follows:

1) Check all the chemicals and materials mentioned in the manuscript and make sure all of the commercial reagents have been labeled as ( Supplier, City, Country). Most of the reagents the authors mentioned are labeled properly. But some of the reagents do not have full information. For example, Line301, it should be (pGL-2, Promega, Wisconsin, USA).

2) Regarding the specific steps of SSV: why use such an old article, it is still about the vitrification method of ovarian tissue instead of testis tissue, because there are many new articles about SSV of testicular tissue. In short, the materials and methods of SSV are not written rigorously enough.

If it is possible, please provide the supplementary video or detailed figures for better understanding of the modified SSV protocols.

5. The entire article requires moderate English language and style editing.

6. This article has fewer citations, and more citations are suggested.

7. It is recommended to rewrite the discussion and please refer to the detailed comments No.7-9 below.

8. In conclusion part, the sentence on Line 403-405 “Our results are in agreement with the review article …” should be moved into discussion part. This citation and the subsequent thinking also can be expanded in discussion as a paragraph. But the conclusion should be original and summative sentences without citation.

Detailed Comments:

1 For figure 1, it is recommended to rewrite LN to LN2.

2 For line 133, it is recommended to rewrite red to yellow,rewrite background control to negative control. The corresponding part of figure 3 should also be rewritten.

3 It is recommended to mark the number of groups of fresh tissue, with CPA and without CPA

4 For line 131, why fresh ITT is 1 µL and SSV ITT is 5 µL?

5 For line 288, It is not recommended to directly quote 6 articles in one sentence.

6. On Line 347, why did the authors choose 31 days as a time point for detecting and evaluation of the ITT? Regarding sentence on Line347 We monitored all the transplanted grafts..., here should be more information about 31days until adulthood.

7. Dicussion paragraph on Line256-272, it is regardingPolII-luc transgenic animal model. This part should be described after the paragraph on Line214-229, which means the PolII-luc transgenic animal model should be the third paragraph of discussion.

Regarding this part, authors mentioned their previous article [25], it showed the peak of spermatogenesis occurred between the 42nd and 55th days in the scaffold group. However, in this manuscript, authors used the same animal model with different transplanted grafts only for 31days culture, shorter than their previous culturing duration. Is there any information available after 31days? For example, what happened after 31days culture of the grafts? Are they died?

8. Line 230-255, in the paragraph authors summarized the studies regarding ITT cryo-storage. It is a really interesting paragraph and it is suggested to add a table or a summary figure for better understanding and clearer presentation of authors idea. And this paragraph should be placed before Limitation of this study.

9. Line 279-283, authors cited the two study[32], [6] to showed livebirth after ITT in vitro and in vivo culture. However, there are more studies regarding livebirth after ITT transplantation, and it is also an interesting and valuable topic. For example, Generation of live piglets for the first time using sperm retrieved from immature testicular tissue cryopreserved and grafted into nude mice. PLoS One. 2013 Jul 29;8(7):e70989. doi: 10.1371/journal.pone.0070989. PMID: 23923039.

Authors could summarize the studies regarding ITT livebirth and write some comments in discussion.

Author Response

Response to reviewer 1

General improvement:

1 The introductory part is too long, especially on Lines 42 and 79, the two sentences are probably the same meaning. It is recommended that the author shorten the whole introductory part. And in line 67-78, the author has quoted his 5 articles in this small paragraph, so it is recommended to reduce some of them.

According to your comment, we changed line 79 to line 75-76, “Testicular cryopreservation is still considered an experimental procedure and the protocols vary between studies.” We also reduce some of the articles quoted and rewrite the paragraph as line 67-74, “The development of noninvasive bioluminescence imaging (BLI) makes it possible to follow tissue-specific luciferase expression in transgenic mice and monitor biological processes such as signaling or protein–protein interactions of transplanted tissues in vivo [19]. This technology has been applied in our experiments to track germ cells, ovarian tissues, and ITTs for over a dozen years. As described in our previous serial studies[20, 21], we used FVB/N-Tg (PolII-luc) Ltc transgenic mice and showed that BLI is a viable tool to track the survival of ovarian grafts. Later, BLI was introduced as a tool to evaluate germ cells and ITT in pre-pubertal male mice [22, 23].”

2 The grouping of this article is still a bit confusing. Why are SSV group in Figure 3/5 parts divided into two groups (with CPA and without CPA), but not in Figure 4/6/7 parts?

We appreciated your valuable opinion. In figure 3 and 5, we confirmed that the ITTs in the group of SSV without CPA is unlikely to be viable. Therefore, we did not continue this group in the further study of IHC stain and in vivo BLI image evaluation. In order to make the article more clearly, we changed the sequencing of these figures. We will first describe in vitro BLI image and SEM results (figure 3, 4), and then demonstrate the results of IHC stain, in vivo BLI and H&E stain (figure 5, 6, 7).

3 About Figure 6 part, From the 7th to the 31st day, although bioluminescence intensity will increase, it is recommended to measure once or twice during this period.

Thank you for your valuable opinion. In our previous study published in Int. J. Mol. Sci., we have demonstrated that the BLI intensity of testicular grafts gradually increase on the 5th to 42nd day after transplantation. This trend should be applied to this study as well. However, we agreed with you and added in our limitations in the discussion, please referred to line 298-300, “ We only evaluated in vivo BLI on day 1, 2, 5, 7 and 31, and a longer follow up duration and more frequent data collections may give us more information about the BLI trend of in vivo grafts.”

4 It is recommended to revise the methods and materials part as follows:

1) Check all the chemicals and materials mentioned in the manuscript and make sure all of the commercial reagents have been labeled as (Supplier, City, Country). Most of the reagents the authors mentioned are labeled properly. But some of the reagents do not have full information. For example, Line301, it should be (pGL-2, Promega, Wisconsin, USA).

Thank you for your opinion, and we have checked the label for each reagent.

2) Regarding the specific steps of SSV: why use such an old article, it is still about the vitrification method of ovarian tissue instead of testis tissue, because there are many new articles about SSV of testicular tissue. In short, the materials and methods of SSV are not written rigorously enough.

If it is possible, please provide the supplementary video or detailed figures for better understanding of the modified SSV protocols. 

We appreciated your valuable opinion. We added more new articles about SSV of ITT and rewrote some of the paragraph to make the description more clearly (Please refer to line 339-348).

  1. The entire article requires moderate English language and styleediting.

Thank you for your opinion. This manuscript has been carefully revised by Online English to improve the grammar and readability. If you consider necessary, we would request for an English editing service again from MDPI.

  1. This article has fewer citations, and more citations are suggested.

We deleted some old references and added some more recent citations for your references. There are now a total of 47 references.

  1. It is recommended to rewrite the discussion and please refer to the detailed comments No.7-9 below.

We rewrote the discussion according to your opinions. Please refer to No. 7-9 below. We also added a new table to summarize studies for ITT cryopreservation.

  1. In conclusion part, the sentence on Line 403-405 “Our results are in agreement with the review article …” should be moved into discussion part. This citation and the subsequent thinking also can be expanded in discussion as a paragraph. But the conclusion should be original and summative sentences without citation.

We have rewritten the conclusion accordingly at line 422-427, “Based on our surrogate mouse model, the in vivo BLI intensities of SSV and fresh ITT grafts are similar, indicating that SSV ITT grafts can survive, accompanied by spermatogenesis, until adulthood. This model may serve as a platform for future versatile study designs and applications from bench to bedside for pre-pubertal male fertility preservation.”

Detailed Comments:

For figure 1, it is recommended to rewrite LN to LN2.

Thank you for your opinion, and we modified the figure accordingly.

2 For line 133, it is recommended to rewrite red to yellow,rewrite background control to negative control. The corresponding part of figure 3 should also be rewritten.

3 It is recommended to mark the number of groups of fresh tissue, with CPA and without CPA

4 For line 131, why fresh ITT is 1 µL and SSV ITT is 5 µL?

We appreciated your valuable opinions. We modified the figure accordingly and we labled each group as following: the first group is DPBS buffer only without ITT tissues, the second group is 1 µL of fresh ITT, the third group is negative control, the forth group is 5 µL vitrified ITT without CPA and the fifth group is 5 µL vitrified ITT with CPA. We hope that the labeling for each group could improve the readibility.

We explain the reason for 1 µL fresh ITT and 5 µL vitrified ITT respectively at line 129-132, “We only added 1 µL, instead of 5 µL, of fresh ITT because of the limitation of the BLI image system. The QYs for 5 µL of fresh ITT is too strong to be analyzed by this system”.

5 For line 288, It is not recommended to directly quote 6 articles in one sentence.

According to your comment, we deleted two of them and only kept recent articles for the readers’ references.

  1. On Line 347, why did the authors choose 31 days as a time point for detecting and evaluation of the ITT? Regarding sentence on Line347 We monitored all the transplanted grafts..., here should be more information about 31days until adulthood.

We have designed this experiemnt based on previous study on rodent testicular development. According to your opinion, we rewrote and explained our experimental design in line 365-370, “ We monitored all the transplanted grafts in vivo using BLI technology for 31 days. According to previous study [32], spermatogenesis begins since 30th postnatal day (PND). The mice should reach its puberty during 35th to 55th PND depending on different strain. Our donor mice are 21st PND and we could expect that the steady and rapid spermatogenesis should begin after 31 days of transplantation (52nd PND).”

  1. Dicussion paragraph on Line256-272, it is regardingPolII-luc transgenic animal model. This part should be described after the paragraph on Line214-229, which means the PolII-luc transgenic animal modelshould be the third paragraph of discussion.

Thank you for your opinion. We changed the original Line 256-272 to Line 223-239, and it is now the third paragraph of discussion.

Regarding this part, authors mentioned their previous article [25], it showed the peak of spermatogenesis occurred between the 42nd and 55th days in the scaffold group.” However, in this manuscript, authors used the same animal model with different transplanted grafts only for 31days culture, shorter than their previous culturing duration. Is there any information available after 31days? For example, what happened after 31days culture of the grafts? Are they died?

Thank you for your opinion. We have examined the in vitro histology after transplantation and presented our result in figure 7. According to our findings, the ITT in both SSV and fresh groups preserve intact structures for spermatogenesis on the 31th day after transplantation, and mature sperm was found. We agreed that a longer follow up duration may give us more information about the BLI trend of in vivo grafts, and we add this limitation in our discussion (line 298-300).

  1. Line 230-255, in the paragraph authors summarized the studies regarding ITT cryo-storage. It is a really interesting paragraph and it is suggested to add a table or a summary figure for better understanding and clearer presentation of authors’ idea. And this paragraph should be placed before Limitation of this study.

 Thank you for your valuable opinion and we summarized the experimental method and conclusion of each article and add a new table (Line 270-272).

  1. Line 279-283, authors cited the two study[32], [6] to showed livebirth after ITT in vitro and in vivo culture. However, there are more studies regarding livebirth after ITT transplantation, and it is also an interesting and valuable topic. For example, Generation of live piglets for the first time using sperm retrieved from immature testicular tissue cryopreserved and grafted into nude mice. PLoS One. 2013 Jul 29;8(7):e70989. doi: 10.1371/journal.pone.0070989. PMID: 23923039.

Authors could summarize the studies regarding ITT livebirth” and write some comments in discussion.

We have added the reference you suggested and rewrote a new paragraph discussing live birth after ITT cryopreservation. We also mentioned live birth after SSCs cryopreservation and in vitro culture. There are scanty studies reported live birth after ITT cryopreservation. In 2002, using sperm developed in the frozen–thawed transplants, the first live mouse offspring was born after in-vitro micro-insemination [37]. Another more recent study retrieved post-meiotic spermatids 8–12 months after transplantation and successfully produced offspring from autologous grafting of cryopreserved pre-pubertal rhesus monkeys [6]. Both studies adopted slow freezing for ITT cryopreservation. Kaneko et al. successfully generated porcine offspring utilizing sperm from immature testicular tissues after vitrification and transplantation into nude mice [38]. Testicular xenografting animal was adopted in this study. There are some ethical and methodological obstacles that need to be overcome before this technology could be applied to human beings. In addition, in vitro culture is an interesting method to avoid re-introduction of malignant cells to cancer patients, especially with haematological malignancies [39]. Sato et al. were the first group to show complete spermatogenesis from cryopreserved ITT after they were thawed and culture in vitro. Offspring was generated from spermatids and sperm produced in vitro by micro-insemination [40]. Since then, cryopreservation of a mixed population of testicular cell suspensions, enriched in SSCs, has been envisaged as a simple and cost-effective alternative to preserve male fertility. There are some recent studies investigated the optimal method for SSCs preservation, including SSV [41, 42]. To the best of our knowledge, there is currently no live birth of human after either frozen-thawed ITT transplantation or in vitro culture. (line 273-291)

Reviewer 2 Report

I read with interest this manuscript evaluating the trasplantation of immature testicular tissue in donor mice comparing fresh and solice-surface vetrified tissue. the manuscript is clear and well writtten. The study design is detailed and the results corresponding to the aim of the paper.
This study represents an original model for fertility preservation in pre-pubertal male as no alternative are now available for this cathegory of patients.

The current study represent a significant model for fertility preservation in prepubertal male patients. And A detailed analysis of the safety of grafted testicular tissue even after vitrification

e aimed to evaluate in vitro and in vivo bioluminescence imaging (BLI) for solid surface-vitrified (SSV) ITT grafts until adulthood. The donors and recipients were transgenic and wild-type mice, respectively, with fresh ITT grafts used as the control group. No additional controls are necessary. Detailed descriptions about tables and figures are already provided in the manuscript. And the references are appropriate.

Author Response

Response to reviewer 2:

I read with interest this manuscript evaluating the transplantation of immature testicular tissue in donor mice comparing fresh and solid-surface vitrified tissue. the manuscript is clear and well written. The study design is detailed and the results corresponding to the aim of the paper. The current study represents a significant model for fertility preservation in prepubertal male patients aimed to evaluate in vitro and in vivo bioluminescence imaging (BLI) for solid surface-vitrified (SSV) ITT grafts until adulthood. The donors and recipients were transgenic and wild-type mice, respectively, with fresh ITT grafts used as the control group.

Answer

We appreciate your valuable opinion about our manuscript. You pointed out that our study represents a novel mouse model for future research in the field of pre-pubertal male fertility preservation. We hope that there will be more studies adopting this transgenic mouse model to improve the technology of ITT cryopreservation in the future.
